# Recent Update and Drug Target in Molecular and Pharmacological Insights into Autophagy Modulation in Cancer Treatment and Future Progress

**DOI:** 10.3390/cells12030458

**Published:** 2023-01-31

**Authors:** Md. Ataur Rahman, Abu Saim Mohammad Saikat, Md. Saidur Rahman, Mobinul Islam, Md. Anowar Khasru Parvez, Bonglee Kim

**Affiliations:** 1Department of Pathology, College of Korean Medicine, Kyung Hee University, 1-5 Hoegidong Dongdaemun-gu, Seoul 02447, Republic of Korea; 2Korean Medicine-Based Drug Repositioning Cancer Research Center, College of Korean Medicine, Kyung Hee University, Seoul 02447, Republic of Korea; 3Department of Biochemistry and Molecular Biology, Life Science Faculty, Bangabandhu Sheikh Mujibur Rahman Science and Technology University, Gopalganj 8100, Bangladesh; 4Department of Animal Science & Technology and BET Research Institute, Chung-Ang University, Anseong 17546, Republic of Korea; 5Department of Energy and Materials Engineering, Dongguk University, Seoul 04620, Republic of Korea; 6Department of Microbiology, Jahangirnagar University, Savar, Dhaka 1342, Bangladesh

**Keywords:** autophagy, cancer, autophagy inhibitors, autophagy activators, autophagic cell death

## Abstract

Recent evidence suggests that autophagy is a governed catabolic framework enabling the recycling of nutrients from injured organelles and other cellular constituents via a lysosomal breakdown. This mechanism has been associated with the development of various pathologic conditions, including cancer and neurological disorders; however, recently updated studies have indicated that autophagy plays a dual role in cancer, acting as a cytoprotective or cytotoxic mechanism. Numerous preclinical and clinical investigations have shown that inhibiting autophagy enhances an anticancer medicine’s effectiveness in various malignancies. Autophagy antagonists, including chloroquine and hydroxychloroquine, have previously been authorized in clinical trials, encouraging the development of medication-combination therapies targeting the autophagic processes for cancer. In this review, we provide an update on the recent research examining the anticancer efficacy of combining drugs that activate cytoprotective autophagy with autophagy inhibitors. Additionally, we highlight the difficulties and progress toward using cytoprotective autophagy targeting as a cancer treatment strategy. Importantly, we must enable the use of suitable autophagy inhibitors and coadministration delivery systems in conjunction with anticancer agents. Therefore, this review briefly summarizes the general molecular process behind autophagy and its bifunctional role that is important in cancer suppression and in encouraging tumor growth and resistance to chemotherapy and metastasis regulation. We then emphasize how autophagy and cancer cells interacting with one another is a promising therapeutic target in cancer treatment.

## 1. Introduction

Recent evidence indicates that autophagy recycles misfolded proteins and damaged organelles, maintaining cellular homeostasis in neurodegeneration and cancer. It includes a sequence of carefully regulated/controlled processes (i.e., initiation, nucleation, elongation, lysosomal fusion, and destruction) [1] (Figure 1). Research over the previous two decades has highlighted autophagy’s role in human diseases [2]. As a key evolutionary catabolic process for cytoplasmatic component digestion, altering the autophagic activity by targeting certain regulatory factors may affect disease processes [3]. Autophagy’s activation and downregulation in malignancies indicate oncogenic and tumor-suppressive characteristics [4]. Consequently, key autophagy targets must be identified for novel therapeutics.

Autophagy is a dynamic and physiological strategy for stabilizing metabolic equilibrium that relies on the formation of double-membrane vesicles (lysosomes) that encapsulate intracellular components such as obsolete or superfluous proteins and damaged or aged organelles [5]. Because lysosome-released breakdown products are reintroduced into biosynthetic and metabolic processes, basal autophagy is generally considered a cell durability mechanism under nutrient insufficiency. Additionally, autophagy functions as a cytoprotective process against various extrinsic stresses, including oxidative stress (OS), bacterial or viral infection, and endoplasmic reticulum stress (ERS), by removing injured and toxic products and cellular materials [6]. Nevertheless, autophagy has a dual effect in cancer. For example, it inhibits and promotes tumor progression, initiation, growth, and therapy by limiting the toxic buildup of oncogenic signaling molecules caused by carcinogenic chromosomal damage [1]. Cancer cells often use autophagy-mediated recyclable biomolecules to satisfy their higher metabolic energy requirements and to withstand microenvironmental stress, promoting aggressiveness and tumorigenesis [7]. Cancer cells are more autophagy-dependent than normal tissues; and consequently, specifically targeting autophagy is a potential cancer treatment strategy.

Autophagy may serve as a pro-survival or pro-death process to prevent or moderate anticancer drugs’ cytotoxicity. At the molecular level, this mechanism is closely controlled by the autophagy-related gene (ATG) protein family [8]; however, autophagy’s involvement in cancer remains controversial. Depending on the cellular environment, cytotoxic or cytoprotective effects have been described. Therefore, a deeper interpretation of autophagy inspection and its involvement in each cellular environment is critical for selecting a suitable autophagy-modulating therapeutic strategy in cancer.

Numerous treatments currently in preclinical and clinical trials, and many currently approved medications, induce cytoprotective autophagy, including kinase inhibitors, mammalian target of rapamycin (mTOR) inhibitors, antiangiogenic agents, and natural products [9,10,11]. Moreover, several signaling mechanisms and molecules involved in controlling drug-induced autophagy have been identified, including the phosphoinositide 3-kinase (PI3K)-protein kinase B (Akt)-mTOR pathway, one primary controller of autophagy [12,13,14]. Ongoing clinical trials alter autophagy to treat cancer, combining autophagy-affecting substances with anticancer medications targeting the cell signaling pathways, metabolic processes, and others to enhance cancer therapy [15]. Several studies have shown that autophagy is a potential cancer therapy target [16,17]. Autophagy modulation in response to medicines can be pro-death or pro-survival, contributing to an anticancer effectiveness or drug resistance [18]. Therefore, this review presents the current understanding of autophagy modulation, which may help us design a potential therapeutic strategy to enhance chemotherapy effects and improve cancer patient outcomes with an improved therapeutic efficacy.

## 2. Recent Update on the Bifunctional Role of Autophagy-Mediated Cancer Cell Death

Recently, autophagy’s dual function has been targeted to disrupt or eliminate cancer metabolism for medicinal and research purposes [19,20]. Autophagy removes damaged cellular components and proteins and maintains cellular homeostasis to suppress tumors [21]. Additionally, several studies have shown that autophagy promotes advanced cancer survival and growth. Several tumors undergo autophagic cell death when certain anticancer medicines induce autophagy [7]; therefore, inducing autophagy may be an effective treatment approach for certain cancers [22]. Additionally, basal autophagy is considered a cancer-suppressing factor, and several progressed cancer forms have a more basal autophagic operation than normal tissues, including pancreatic cancer or triggered Ras tumors, which have been defined as autophagy-dependent tumors [23].

Autophagy is a well-characterized viability strategy in various tumor forms. This association might be because it protects against nutritional deficiency and substrates fundamental for cell viability; however, it may also be connected to its protective effect against programmed cell death, including apoptosis. Autophagy involvement in regulating and modulating apoptosis and apoptotic checkpoints is critical for developing cancer treatments [5]. The molecular interactions between apoptosis and autophagy in cancer cell treatment are presented in Figure 2.

It is well established that autophagy interacts with and regulates canonical apoptosis, and it has been hypothesized that autophagy regulates apoptosis and vice versa. Consequently, researchers have proposed that a cell may overcome brief periods of autophagy suppression by raising the FOXO3a levels to trigger downstream autophagy targets. Conversely, when autophagy is inhibited for an extended period, pro-apoptotic genes, including PUMA, are activated, sensitizing the cell to death [24]. Numerous proteins that interact with BECN1 function as tumor suppressors and positively regulate autophagy, including the UV radiation resistance-associated gene (*UVRAG*) and Bax interacting factor-1 (*BIF1*) [25]. In colon, gastric, breast, and prostate cancers, UVRAG and BIF1 depletion impaired the autophagosome formation and autophagy, increasing the cancer cell proliferation [26].

Additionally, autophagy-deficient cells showed metabolic stress in animal studies, impairing cell survival; therefore, autophagy helps tumor cells survive by improving their ability to withstand stress and supplying them with nutrients to meet their metabolic needs. Tumor cell death can also result from inhibiting autophagy or knocking down autophagy-related genes [27]. Therefore, these observations suggest that autophagy plays an important bifunctional role in cancer cell promotion and suppression in several tumors.

## 3. Recently Used Drug Targets and Their Interaction with Autophagy to Manipulate Cancer

It has recently been found that autophagy regulation is an intriguing and potentially useful strategy to improve cancer treatments [28,29]. Numerous studies have indicated autophagy’s dual roles for several drugs that might be modulated to suppress or promote tumor growth, and the following sections describe autophagy’s dual function in recent drug treatments.

### 3.1. Targeted Drug for Autophagy Activation in Cancer Therapy

In cancer, autophagy helps promote and inhibit tumors by helping or hindering cancer cells from growing and dividing, while some drugs that fight cancer control autophagy; therefore, chemotherapy controlled by autophagy can cause cancer cells to live or die [30]. Autophagy regulation also affects the expression of proteins that promote or inhibit tumor growth [31], and the mechanism of action associated with autophagy stimulation is presented in Table 1. Additionally, the stimulation of autophagy by various chemicals and medications in cancer treatments is presented in Figure 3.

Autophagy may be tumor-suppressive, neutral, and tumor-promoting in cancer, depending on the microenvironment stress, nutrient availability, immune system, and pathogens [21]. The mTOR protein kinase is related to various biological functions, such as cell survival, growth, immunity, and metabolism. Consequently, mTOR is implicated in supervising numerous biological processes, notably autophagy, apoptosis, and the cell cycle, thus preventing the commencement of the concluding process [44]. A P75NTR activation by rapamycin induces autophagy, leading to Kaposi’s sarcoma cell death [43]. Additionally, rapamycin (sirolimus), a secondary metabolite derived from *Streptomyces hygroscopicus*, has significant anticancer and immunosuppressive activities [45]. Rapamycin and its semi-synthetic counterparts (rapalogs) are allosterically-precise and potent mammalian targets of rapamycin complex 1 (mTORC1) blockers that influence downstream targets, such as autophagy stimulation [46]. Other mTOR blockers interfere with adenosine triphosphate (ATP), preventing the phosphorylation of its corresponding proteins and leading to more effective mTOR suppression [47]. Altogether, these observations show that mTOR blockers may be used to promote cell death through several pathways. Their mechanisms of action rely on the tumor microenvironment, overcoming cancer cell resistance in combination therapy [48].

While cannabinoids have shown strong anticancer properties associated with autophagy, they have also shown cytoprotective properties, depending on the cell type and cannabinoid used [49]. Active cell death (ACD) in melanoma and glioma cells occurs via mTORC1 inhibition and autolysosome permeabilization, resulting in cathepsin secretion and apoptosis activation [50]. JWH-015, for example, is a synthetic cannabinoid specific to the CB2 receptor that suppressed tumor development in hepatocellular carcinoma (HCC) cells in vivo via inhibiting the Akt/mTORC1 cascade through an AMP-activated protein kinase (AMPK) stimulation [9]. Moreover, obatoclax (GX15-070) is another BH3 mimic that induced necroptosis in oral squamous cell carcinoma, acute lymphoblastic leukemia, and rhabdomyosarcoma via autophagic signaling [51]. Additionally, obatoclax promoted autophagy in adenoid cystic carcinoma cells and inhibited autophagy in colorectal cancer cells in the absence of Bcl-1 [52]. Subsequently, ABT-737 was found to be effective in vitro against HCC cells via an autophagy-dependent mechanism involving Bcl-1 [53].

Furthermore, APO866 inhibited nicotinamide adenine dinucleotide biosynthesis and induced apoptotic cell death in hematological cancer cells [54]. Elsewhere, histone deacetylase (HDAC) inhibitors were shown to induce autophagy. A recent study used the HDAC inhibitor, suberoylanilide hydroxamic acid (SAHA), to treat cutaneous T-cell lymphoma and glioblastoma [55]. While apoptosis had been identified as the primary mechanism through which the HDAC inhibitors promoted cancer cell death, autophagy promotion was also associated with a PI3K/Akt/mTOR pathway inhibition [56]. In addition, SAHA inhibited breast cancer cell proliferation in vitro by stimulating autophagy and activating cathepsin B [57,58]. Moreover, in vivo and in vitro studies have shown that MHY2256 (a synthetic class III HDAC inhibitor) stimulates CCA, ACD, and apoptosis in endocervical curettage [59]. SAHA activates autophagy by inhibiting mTOR and upregulating the *LC3* expression. Autophagy mitigates a SAHA-induced apoptotic and nonapoptotic cell death, suggesting that targeting autophagy may improve SAHA’s therapeutic effects [60].

### 3.2. Targeted Drug for the Inhibition of Autophagy in Cancer Therapy

It has been shown that inhibiting autophagy makes an anticancer therapy more effective, suggesting that it is a potentially valuable approach that could be combined with other anticancer therapeutic approaches to improve cancer treatments [61]. Numerous autophagy inhibitors impede the autophagy processes at various stages, as discussed in Table 2 and Figure 4.

Nigrosporins B, a potential anti-cervical cancer agent, induced apoptosis and protective autophagy in human cervical cancer Ca Ski cells. These effects were mediated via the PI3K/AKT/mTOR signaling pathway [62]. Additionally, autophagy is affected by antimetabolites, antiangiogenic agents, DNA-damaging agents, proteasome inhibitors, microtubule-targeted drugs, death receptor agonists, hormonal agents, HDAC inhibitors, and kinase inhibitors [74]. ATG proteins and other factors form complexes that lengthen phagophores. ATG7 is involved in creating the ATG5-ATG12 complex and coupling PE to LC3 and GABARAP [75]. Consequently, many ATG7 suppressors (e.g., WO2018/089786) have been developed, extending the use of micro RNAs (miRNAs) targeting the *ATG7* gene, including miR-154, which suppresses blade cancer growth [76]. Conversely, ATG4B cleaves LC3, stimulating it in preparation for coupling with PE, which is required for autophagosome development and activation [77].

Verteporfin is a photodynamic therapeutic agent derived from benzoporphyrin. It also suppresses autophagosome production induced by serum and glucose deprivation but not mTOR suppression [78]. Verteporfin is suspected of suppressing p62 oligomerization, a protein crucial for ubiquitinated targets to be sequestered into autophagosomes [78]. Inhibiting ULK1 slowed the tumor development and activated apoptosis. This finding has resulted in the discovery of many chemicals that compete for ATP-binding sites, including compound-6, MRT67307, and MRT68921 [9]. Apart from these, the most investigated compound is SBI-0206965, which has been shown to suppress autophagy and promotes apoptosis in neuronal ceroid lipofuscinoses, clear-cell renal cell carcinoma cells, and non-small-cell lung cancer (NSCLC) cells [79]. SF1126, a conjugated LY294002 derivative, was engineered to assemble in integrin-expressing tissues, increasing LY294002′s solubility and pharmacokinetics, promoting tumor site retention, and showing anticancer and antiangiogenic activity in mouse models [80].

3-methyladenine (3-MA) was among the earliest autophagy inhibitors. It inhibits autophagy in starvation situations by inhibiting PI3KC3 [81]. Conversely, when nutrients are present, they induce autophagy by blocking PI3KC1. The VPS34 inhibitor, SAR405, is a (2S)-tetrahydropyrimido-pyrimidinone class member that inhibits kinases via an extreme competition for their ATP binding site, thus suppressing autophagy produced by an mTOR inhibition or malnutrition [82]. VPS34-IN1 is a bipyrimidinamine that preferentially suppresses PI3KC3 [9]. Esomeprazole’s antiproliferative action, for example, was enhanced when combined with the PI3K inhibitor, 3-MA, in gastric cancer cells through EGFR downregulation via the PI3K/FOXO3a pathway [63]. In gemcitabine-resistant pancreatic cancer cells, however, 4’-acetylantroquinonol B promoted cell death and inhibited autophagy by downregulating the PI3K/Akt/MDR1 pathway, increasing cell death [64].

Autophagy’s late phase involves autophagosomes fusing with lysosomes, which contain hydrolases that destroy the autophagosomes’ contents. During this stage, autophagy is suppressed by the lysosomal blockers, bafilomycin A1 (Baf-A1), chloroquine (CQ), and its derivatives Lys05 and hydroxychloroquine (HCQ), which are antimalarial medicines used for treating malaria and, more recently, cancer [50].

Only CQ and HCQ are authorized for therapeutic use as autophagy inhibitors [42]. Consequently, CQ derivatives with increased inhibitory-efficacy against autophagy have been synthesized. Lys05, for example, is a dimeric CQ analog that aggregates better than HCQ within acidic organelles, particularly lysosomes [83]. Duroquinone DQ661 disrupts lysosomal degradation, notably autophagy, and suppresses PPT-1, inhibiting the mTORC1 cascade. Furthermore, DQ661 has shown independent efficacy in tumor mouse models and the ability to overcome gemcitabine resistance [9]. VATG-027 is another antimalaria drug that suppresses autophagy with antitumoral effects [84]. Meanwhile, mefloquine accumulates in lysosomes, impairing autophagy, inducing apoptosis, and inhibiting MDRP1, making it potent against MDRT cells. It sensitizes chronic myeloid leukemia cells created from chronic-stage patients to tyrosine kinase blockers, with a preference for stem/progenitor tumor cells over normal cells [85].

CQ and its analogs, however, are not the only medications that suppress autophagy by affecting lysosomes [86]. Baf A is a vacuolar-H^+^ ATPase blocker that prevents H^+^ from entering lysosomes, vacuoles, and vesicles. Baf A also suppresses autophagosome-lysosome fusion by disrupting the Ca^2+^ gradients required for this pathway [87]. Additionally, ionophores can alter the pH of the lysosome, affecting the autophagy mechanism. Tambjamine derivatives are ASI synthesized from naturally produced tambjamines [88]. Importantly, because lysosomes are involved in tumor invasion, these blockers are efficacious against metastasis by addressing cancer stem cells and promoting tumor vessel normalization.

## 4. Preclinical and Clinical Aspects of Current Autophagy Modulation Therapeutic Applications in Cancer

Current therapeutic autophagy applications may help battle tumor drug resistance. Future research requirements and objectives in many preclinical studies, notably those using in vivo and in vitro models, are combining medicines that iduce defensive autophagy with autophagy suppressors [50]. A phase I study used HCQ with DITZ in patients with ASTs and melanoma. Another phase I/II study evaluated HCQ’s anticancer efficacy in combination with temozolomide and radiation in patients with glioblastoma. The pharmacologics and pharmacokinetics of HCQ’s dose-dependent autophagy suppression have also been investigated [89].

A phase I study found that the proton pump inhibitor, pantoprazole, increased the anticancer effects of doxorubicin (DOX) in patients with ASTs based on preclinical models by increasing its distribution and suppressing autophagy [90]. Additionally, a phase I clinical study examined the pharmacodynamics of combined therapy with HCQ and DOX in companion dogs with spontaneously developing lymphoma [91]. A phase I study used a combination of the autophagy blocker, HCQ, and the proteasome inhibitor, bortezomib, in individuals with relapsed/refractory myeloma [19]. A phase I study examined temsirolimus with HCQ in patients with ASTs and melanoma. Additionally, a phase I trial co-administered HCQ with the HDAC antagonist, vorinostat, in patients with ASTs to assess its influence on tolerability, safety, pharmacodynamics, and pharmacokinetics [10].

Cancer research has shown that autophagy is a suitable target for creating new drugs (Figure 5). The antimalarial medicine, CQ, and its analog, HCQ, are notable autophagy blockers that work by inhibiting the lysosomal compartment. While CQ and HCQ were equally effective in inhibiting autophagy in vitro, their toxicity differed in vivo [92]. Additionally, clinical investigations have shown that CQ cannot entirely suppress autophagy in vivo. These findings indicate that developing more effective autophagy suppressors is crucial for advancing the combination treatment approach into clinical trials. Particular attention should be paid to drugs that selectively inhibit autophagy-related proteins [73,93]. Moreover, additional studies should be conducted to determine the function of autophagy triggered by cancer therapy, the design of favorable coadministration systems for drug delivery, and to develop novel and effective autophagy suppressors.

The US Food and Drug Administration and the European Union authorized everolimus for patients with pancreatic neuroendocrine tumors and advanced hormone receptor-positive and HER2-negative breast cancers in combination with exemestane [94]. Betulinic acid is a pentacyclic triterpenoid produced by various plants that promoted ACD in myeloma cells overexpressing Bcl-2. Resveratrol is a polyphenol molecule found in many plants that decreased BCSL cell growth by inhibiting the WNT/β-catenin signaling cascade [95]. δ-Tocotrienol is one of four vitamin E isomers that showed cytotoxic effects in prostate cancer cells (PCCs) in vitro via an autophagy stimulation induced by ERS [96].

Curcumin is a key component of *Curcuma longa* that induces autophagy. It showed a dual effect, protecting or killing cells depending on the therapy’s timeframe and the dose used [97]. Preclinical research showed that Polyphyllin I (*Paris polyphyllin* rhizoma) had anticancer activity by inducing autophagy in various cancers [98]. Ursolic acid obtained from plants induced an autophagic effect in PTEN-deficient-PC3 PCCs via the Akt/mTOR and Bcl-1 pathways [99]. Paclitaxel is a drug that competes with the natural disintegration of microtubules following cell division, and an acquired sensitivity to paclitaxel, mediated by autophagy, significantly impedes anticancer activities.

By inhibiting autophagy, 3-MA or 2-deoxy-D-glucose may improve the selective toxicity in paclitaxel-resistant HeLa cervical cancer cells [100]. Additionally, blocking autophagy with 3-MA and Baf-A1 enhanced paclitaxel sensitivity in follicular dendritic reticulum cells [101]. Obatoclax may also synergistically enhance paclitaxel-induced apoptosis in bladder cancer via inhibiting the autophagic flux [102]. Moreover, pterostilbene with Baf-A1 or 3-MA enhanced the chemotherapeutic efficacy in chemo-resistant and chemo-sensitive large cell carcinoma (LCC) and triple-negative breast cancer (TNBC) [103]. Moreover, Baf-A1 enhanced chaetocin’s anticancer potential [104].

By competing with autophagy, CQ enhanced topotecan’s cytotoxicity in A549 LCC cells and synergistically enhanced cucurbitacin I’s anticancer efficacy in glioblastoma cells [105]. In addition, 3-MA increased the cell death in BE (2)-C HNC following sulforaphane therapy by inhibiting autophagy [106]. Honokiol-induced cell death was triggered by CQ autophagy suppression, resulting in enhanced anticancer activity in human NSCLC cells [107]. CA-4 is a medication derived from *Combretum caffrum* that has been used in clinical studies to treat solid tumors for over a decade. Autophagy suppression through autophagy blockers (i.e., Baf-A1 and 3-MA), JNK inhibitors, or the Bcl-2 blocker, ABT-737, may enhance CA-4-induced apoptosis [101].

Additionally, autophagy activators include hunger inducers, ERS inducers, rapamycin and its derivatives, small molecule rapamycin enhancers, trehalose, and class I PI3K inhibitors [108]. Many are in preclinical or clinical testing, and some are clinically approved. For example, cisplatin (CDDP) conjugated with CQ or 3-MA increased the chemotherapeutic susceptibility of various malignancies. A CQ and CDDP coadministration effectively re-sensitized CDDP-resistant EC109/CDDP cells by reversing the inhibition of mTORC1 activity-mediated autophagy [109]. 4-acetylantroquinonol B also acts as an autophagy suppressor via the PI3K/Akt/mTOR/p70S6K signaling cascade by inhibiting the autophagic flux, thus increasing the susceptibility of extremely invasive epithelial carcinoma to CDDP [64].

Another platinum-based anticancer treatment, oxaliplatin, induces drug resistance in colorectal cancer cells through the mitogen-activated protein kinase (MEK)/extracellular signal-related kinase (ERK) signaling pathway and HMGB1-mediated autophagy, which is restored by 3-MA [110]. Azithromycin is a macrolide antibiotic used to treat many bacterial infections. It was found to act as an autophagy inhibitor in cystic fibrosis patients by inhibiting lysosomal acidification and autophagy [32]. Elsewhere, mTOR was found to govern cell growth, survival, proliferation, and autophagy in mammals. mTOR dysregulation is associated with cancer, and neurological, cardiovascular, and renal disorders, making it an excellent therapeutic target [111]. Most autophagy inhibitors, however, are still in preclinical development; therefore, their use in therapeutic regimens requires further experimental studies.

## 5. Limitations and Recent Updates of Autophagy-Mediated Strategies for Cancer Therapy

The fact that autophagy plays two different roles in tumors has led to the development of two different ways to fight cancer: inhibition and promotion, and the situation must be considered when choosing the best plan. Due to tumor cell protection and the lack of specific autophagy regulation strategies, autophagy modulation is a major cancer treatment; however, because tumors have defense mechanisms, such as multidrug resistance, tumor immune escape, and even autophagy protection, appropriate autophagy modulation has become crucial in treating cancer [112]. Currently, using autophagy-related drugs is limited by their non-specificity and off-target effects, and there are currently no methods to control autophagy. Currently, autophagy can be turned on or off by rapamycin, CQ, HCQ, and a few other drugs approved for human use, but they were not designed for this purpose [113]. There are also some problems that make it hard to design regulators of clinical autophagy. Recently, nanoparticles (NPs) and miRNAs have shown unique properties that can help to overcome these limitations.

NPs targeting autophagy are under consideration for cancer treatment [114]. NPs can deliver autophagy regulators and chemotherapeutic agents for synergistic cancer therapy. Gold and zinc oxide NPs can also enhance OS-mediated apoptosis and autophagy, reducing cancer progression [1]. Collagen, gelatin, PLGA, and albumin are used to make polymeric NPs. The chemotherapeutic agent, DOX, and the autophagy inhibitor, LY294002, were encapsulated in polymeric NPs to penetrate Poral cancer cells [115]. Additionally, chitosan NPs also influence autophagy in cancer. For example, HeLa and SMMC-7721 cells exposed to chitosan NPs overproduced reactive oxygen species (ROS) and underwent autophagy. Moreover, chitosan-based NPs have been shown to initiate autophagy by increasing LC3 accumulation [116]. By activating autophagy, the chitosan/polycaprolactone NPs enhanced the anti-neoplastic efficacy of 5-fluorouracil in patients with head and neck cancer [117].

Gold-loaded polymeric NPs blocked autophagosome-lysosome fusion by dissociating micelles and damaging the lysosomes. Gold inhibits thioredoxin reductase and increases ROS, inducing autophagy and cell death in breast cancers [118]. Hyaluronic acid-modified shell-core NPs loaded with RIP3 conjugated to CQ and eliminated colon cancer cells via programmed cell death pathways [119]. Black phosphorus quantum dots (QDs) disguised with a platelet membrane provided Hederagenin (HED) in breast tumor therapy (MCF-7 cells), and these HED-loaded QDs increased Beclin-1 and LC3-II to enhance the HED’s antitumor efficacy against breast cancer [120].

Iron NPs accumulated in the mitochondria of colorectal cancer cells, disrupting the glucose metabolism and inducing autophagy [121]. Recent research has shown how silver NPs can regulate autophagy to prevent cancer spread. The nuclear factor, kappa B, regulates autophagy in colorectal tumors [122]. Selenium NPs coupled with astragalus polysaccharides suppressed autophagy and promoted apoptosis in MCF-7 cells [123]. By suppressing autophagy, solid lipid NPs augmented rutin’s protective effect against the neurotoxicity induced by streptozotocin in PC12 cells [124]. Baicalin-loaded folic acid-modified albumin NPs induced autophagy via the ROS-mediated p38 MAPK and Akt/mTOR pathways in MCF-7 cells [125]. Consequently, advanced nanocarriers have been shown to increase the potential for suppressing cancer and regulating autophagy.

Additionally, autophagy can be controlled by genetic tools targeting autophagy-regulating molecules, including miRNAs, the CRISPR/Cas9 system, short-hairpin RNAs (shRNAs), and small interfering RNAs (siRNAs). Exosomal miRNA-4535, found in melanoma stem cells, inhibits the autophagy pathway, promoting metastasis [126]. miRNA-99b-5p targets the mTOR/AR pathway to induce autophagy and suppress PCC growth [127]. By regulating the DNMT3B-mediated DNA methylation of RBP1, miRNA-20a-5p can limit autophagy in ovarian cancer patients, decreasing their sensitivity to CDDP [128].

In addition, isorhapontigenin inhibits human bladder cancer invasion via an upstream regulatory Dicer/miR-145/SOX2/miR-365a/RAC1 cascade resulting in MKK7/JNK activation and autophagy induction [129]. Consequently, the simultaneous inhibition of IGF1R, ERK, and autophagy increased the cytotoxicity in pancreatic ductal adenocarcinoma cell (PDAC) lines and reduced the human PDAC organoid survival. A joint IGF1R and ERK inhibition increased the efficiency of autophagy inhibitors in PDAC cells when a CRISPR-Cas9 loss-of-function screen was used [130]. Moreover, shRNA-mediated *FKBP1A* suppression and 3-MA reduced dendrobine’s effects. The dendrobine suppressed inflammation, OS, apoptosis, and senescence in oxidized low-density lipoprotein-treated human umbilical vein endothelial cells [131]. By activating autophagy and targeting the miR-506-3p/FOXP1 axis, the long non-coding RNA XIST was responsible for promoting a carboplatin resistance in ovarian cancer [132]. In addition, metformin may promote ferroptosis by blocking autophagy through the small interfering RNA, H19, potentially making it easier to design innovative medicines for breast cancer [133]. Additionally, leelamine (LEE) inhibits the development of melanoma cells, breast cancer cells, and PCC, and finally, a STAT5 siRNA knockdown enhanced LEE-induced apoptosis, autophagy, and oncogenic protein levels [134].

## 6. Conclusions and Future Directions

Our knowledge of autophagy in cancer is limited. To properly appreciate this complicated and entangled relationship, autophagy must be understood in the context of tumor complexity and heterogeneity. While targeting autophagy appears to be a viable therapeutic method for cancer, considerable difficulties remain to be overcome to enhance the clinical outcomes. Despite autophagy’s paradoxical and contradictory involvement in tumor development and progression, cellular and animal models have shown the major roles of autophagy activities in malignancies. Future approaches may include enhanced medications that modulate autophagy and improved clinical trial designs and patient selection. To successfully adapt the preclinical information to a clinical context, scientists and doctors must better understand the molecular underpinnings of autophagy and carcinogenesis. New autophagic modulators, activators, or inhibitors are needed to target the recently identified autophagic signaling molecules critical in canonical and alternative autophagy pathways. This review has highlighted autophagy’s dual role in cancer, revealing that a better understanding of how autophagy affects cancer development, new biomarkers to delineate the responder patient populations to specific autophagy modulators, and simple and efficient pharmacodynamic indicators to monitor patient responses are urgently needed to improve clinical trials.

## Figures and Tables

**Figure 1 cells-12-00458-f001:**
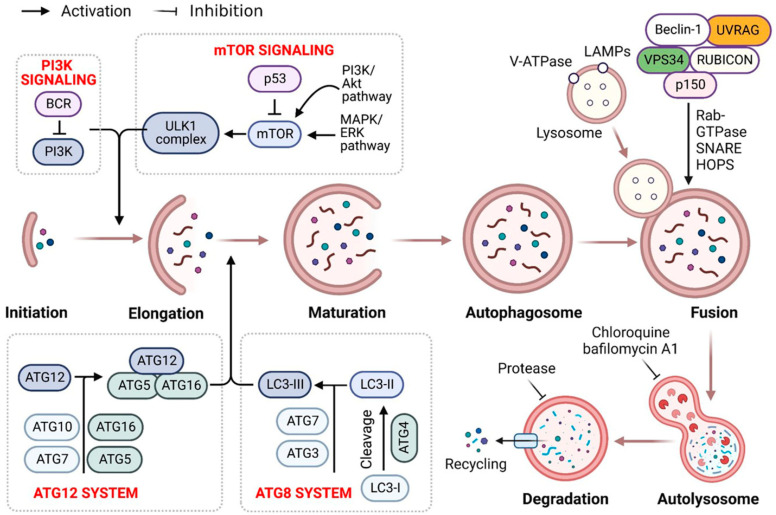
Autophagy signaling at the molecular level. Typically, autophagy is triggered by a shortage of developmental factors or nutrition, which inhibits AMP-activated protein kinase (AMPK) or mTOR. Following Beclin-1 phosphorylation, it activates VPS34, initiating phagophore formation. Conjugation of ATG5-ATG12 includes ATG7 and ATG10, forming an ATG12-ATG5-ATG16 complex that also induces phagophore formation. ATG12 and ATG5 form the ATG16 complex, which serves as an E3 function in the LC3-phosphatidylethanolamine (PE) assembly operation (LC3-II). Similarly, this compound begins phagophore formation. LC3-II is a specific autophagy marker eventually disrupted by autolysosomes. Autophagosome maturation results in its fusion with lysosomes and multiple lysosomal proteins, resulting in cargo destruction and metabolite and nutrient recycling.

**Figure 2 cells-12-00458-f002:**
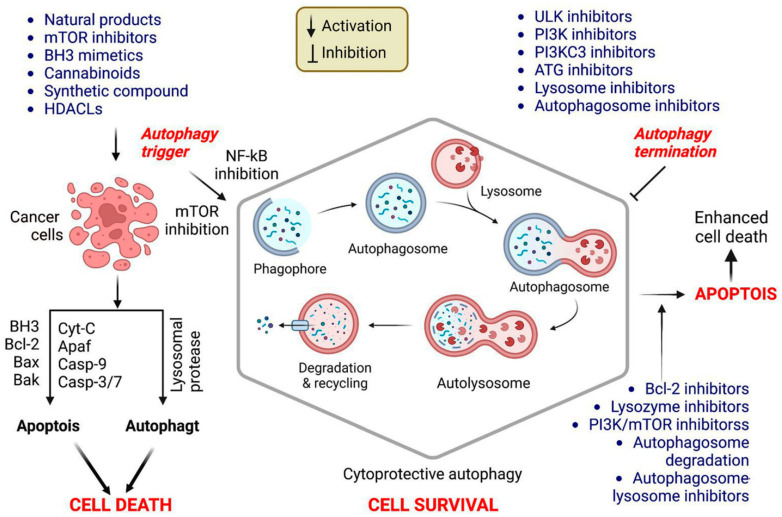
Mechanism of autophagy promotion and repression in cancer therapy. Apoptosis requires enhanced outer mitochondrial membrane permeability, which is mediated by the B-cell lymphoma 2 (Bcl-2) family (e.g., multidomain pro-apoptotic members Bak and Bax), culminating in the translocation of cytochrome (Cyt) C into the cytoplasm. Then, Cyt C binds to Apaf-1, activating the caspase pathways and inducing apoptosis. Bcl-2 family members predominantly regulate mitochondrial membrane permeability during apoptosis. When apoptosis is inhibited, various apoptotic triggers induce autophagy, culminating in active cell death (ACD). Cancer cells treated only with autophagy enhancers undergo apoptosis or autophagy or will activate cytoprotective autophagy, resulting in treatment resistance. Combining autophagy inducers and inhibitors enhances the cytotoxicity in cancer cells via cytoprotective autophagy inhibition. Further studies should be conducted to determine the function of autophagy triggered by cancer therapy, to design advantageous coadministration systems for drug delivery, and to uncover novel and effective autophagy suppressors.

**Figure 3 cells-12-00458-f003:**
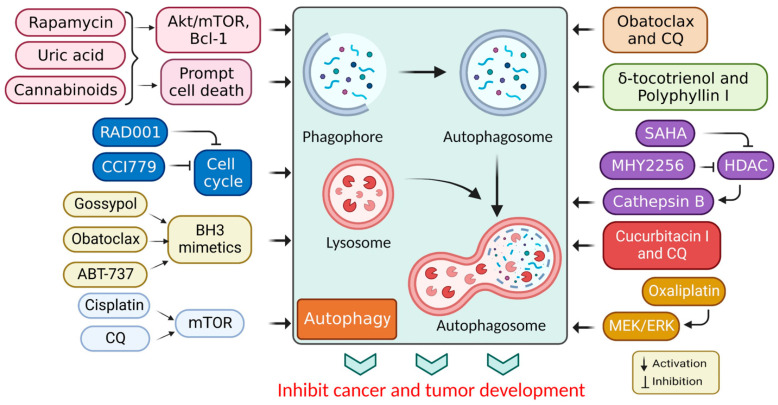
Autophagy stimulation by several compounds and drugs used in cancer therapy.

**Figure 4 cells-12-00458-f004:**
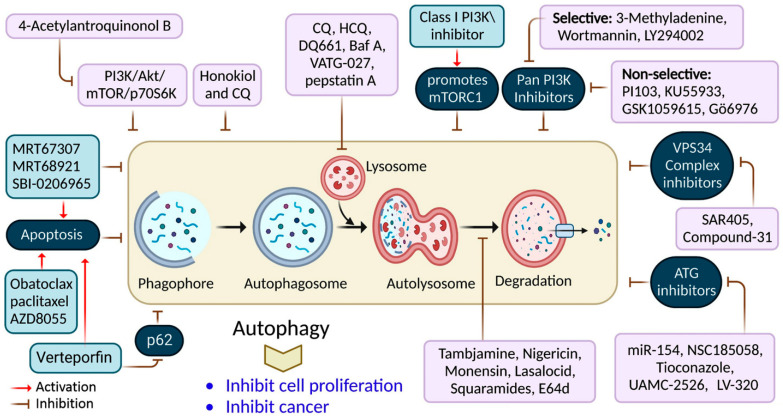
Autophagy inhibition by several compounds and drugs used in cancer therapy.

**Figure 5 cells-12-00458-f005:**
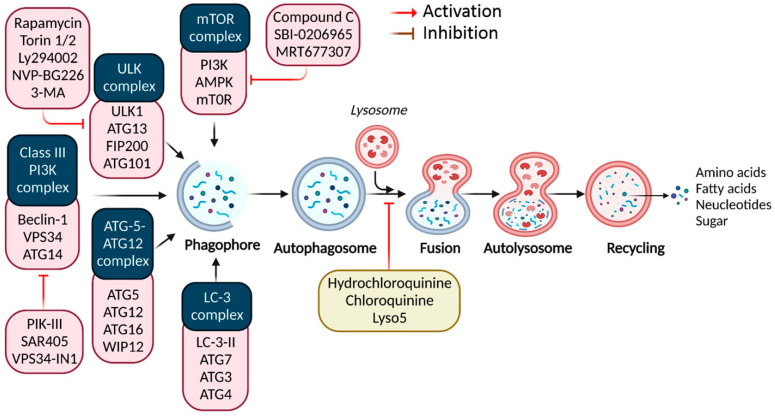
Cellular targets of medications targeting autophagy in cancer in preclinical and clinical settings. Clinical investigations involving autophagy suppression.

**Table 1 cells-12-00458-t001:** Recently used drugs for autophagy stimulation and their mechanisms of action.

Drug	Model	Molecular Mechanism	Reference
Azithromycin	Non-small-cell lung cancer (NSCLC) cells	Lysosomal membrane permeabilization for apoptosis induction	[32]
Penfluridol	Renal cell carcinoma	Induction of autophagy-mediated apoptosis and stemness inhibition	[33]
Canagliflozin	Acute kidney injury (AKI)	Cisplatin-mediated AKI by activating AMPK and autophagy	[34]
Oleanolic acid (3β-hydroxyolean-12-en-28-oic acid)	Different cancer model	Inducing tumor cell apoptosis via activating autophagy	[35]
Epigallocatechin-gallate	Human aortic epithelial cells	Autophagic flux stimulation	[36]
Korean red ginseng	Colon carcinoma cells (HCT-116 and SNU-1033)	Mitochondrial reactive oxygen species (ROS)-mediated autophagy and apoptosis.	[37]
Betulinic acid	Human malignant melanoma cell lines	Inhibits tumor growth and induces autophagy	[38]
Curcumin	Human thyroid cancer cells	Mitogen-activated protein kinase (MAPK) activation and mTOR inhibition with autophagy induction	[39]
Ursolic acid	Human breast cancer cells (MCF-7/MDA-MB-231)	Class III PI3K(VPS34)/Beclin-1 pathway autophagy induction	[40]
Obatoclax	Atypical teratoid/rhabdoid tumors	Dual mammalian target of rapamycin complex 1/2 (mTORC1/2) inhibition	[41]
Chloroquine	Triple-negative breast cancer (TNBC)	PI3K/Akt inhibitors and induces autophagy	[42]
Rapamycin	Kaposi’s sarcoma	PI3K/Akt/mTOR activation autophagy	[43]

**Table 2 cells-12-00458-t002:** Recently used autophagy inhibitors and their mechanisms of action in different cancers.

Drug	Model	Molecular Mechanism	Reference
Nigrosporins B	Human cervical cancer cells (Ca Ski)	PI3K/AKT/mTOR-mediated autophagy inhibition	[62]
3-methyladenine (3-MA)	Gastric cancer cells	PI3K/AKT/FOXO3a inhibition	[63]
4-acetylantroquinonol B	Human pancreatic cancer cells (MiaPaCa-2 and GEM-resistant MiaPaCa-2)	Upregulation of reactive oxygen species (ROS) promoted apoptosis via autophagy inhibition	[64]
LY294002	Laryngeal squamous cell carcinoma	Inhibition of autophagy via the PI3K/mTOR pathway	[65]
MRT67307	Several cancer models and lung cancer cells (A549)	ULK1 inhibition and autophagy inhibition	[66]
Honokiol	Human medulloblastoma	ERK/p38-MAPK-mediated autophagic inhibition	[67]
Obatoclax	Different cancer cells	Mitochondrial Ca^2+^ overload and autophagy inhibition	[68]
MHY2245	Human colorectal cancer cells (HCT116)	Cell cycle arrest and apoptosis	[69]
Hydroxychloroquine	Human pancreatic cancer cells (PANC-1, MiaPaCa-2, and BxPC-3)	Promotes Bcl-xL inhibition-induced apoptosis with autophagy inhibition	[70]
Wortmannin	Pancreatic cancer cells (PANC-1, BxPC-3, and Capan-2)	Nutrient-starvation conditions via autophagy inhibition	[71]
Bafilomycin A1	Thyroid cancer cells (MDA-T32, MDA-T68, FTC-133, and 8505c)	Autophagy inhibition reduced cell migration and invasion	[72]
Tioconazole	Human colorectal cancer cells (HCT116)	Autophagy inhibition	[73]

## Data Availability

Upon request to corresponding author.

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
