# Peer review of "Recent Update and Drug Target in Molecular and Pharmacological Insights into Autophagy Modulation in Cancer Treatment and Future Progress"

_cells, 2023, doi:10.3390/cells12030458_

Round 1

Reviewer 1 Report

The review article entitled, “Recent Update and Drug Target in Molecular and Pharmacological Insights into Autophagy Modulation in Cancer Treatment and Future Progresses” provided a critical analysis and correlation of knowledge about dual roles of autophagy of cancer. Recent studies examining the anticancer efficacy of drugs and practical use of targeting cytoprotective autophagy were updated. The general molecular process of autophagy and the dual role of autophagy in cancer suppression and resistance to treatment were also discussed, In general, this article is clearly presented and the illustrations are appropriate. This review is of high relevance and a pleasure to read. It can be accepted and published

Author Response

The review article entitled, “Recent Update and Drug Target in Molecular and Pharmacological Insights into Autophagy Modulation in Cancer Treatment and Future Progresses” provided a critical analysis and correlation of knowledge about dual roles of autophagy of cancer. Recent studies examining the anticancer efficacy of drugs and practical use of targeting cytoprotective autophagy were updated. The general molecular process of autophagy and the dual role of autophagy in cancer suppression and resistance to treatment were also discussed, In general, this article is clearly presented and the illustrations are appropriate. This review is of high relevance and a pleasure to read. It can be accepted and published

>> (Response) First of all, we would like to express our sincere gratitude for the time and effort the reviewer had put into reviewing our manuscript. We are thankful to the reviewer to accept our manuscript. Additionally, we checked English language of our manuscript to improve the quality of our manuscript (Company name: Cambridge Proofreading LLC, Invoice No: 445-12-40).

Reviewer 2 Report

see comments for editor. Please communicate to authors that they must significantly improve the readability of the manuscript in order for it to be reviewed.

Author Response

see comments for editor. Please communicate to authors that they must significantly improve the readability of the manuscript in order for it to be reviewed.

>> (Response) First of all, we would like to express our sincere gratitude for the time and effort the reviewer had put into reviewing our manuscript. As your suggestion, professional language editors reviewed and improved the quality of our manuscript's English (Company name: Cambridge Proofreading LLC, Invoice No: 445-12-40).

Certificate is attached here:
